# Mental Health Literacy and Positive Mental Health in Adolescents: A Correlational Study

**DOI:** 10.3390/ijerph19138165

**Published:** 2022-07-03

**Authors:** Joana Nobre, António Calha, Henrique Luis, Ana Paula Oliveira, Francisco Monteiro, Carme Ferré-Grau, Carlos Sequeira

**Affiliations:** 1Health School, Polytechnic Institute of Portalegre, 7300-555 Portalegre, Portugal; henrique.luis@ipportalegre.pt (H.L.); paulaoliveira@ipportalegre.pt (A.P.O.); franciscomonteiro@ipportalegre.pt (F.M.); 2Faculty of Nursing, University of Rovira i Virgili, 43003 Tarragona, Spain; carme.ferre@urv.cat; 3Nursing Research Unit for South and Islands (NURSE’IN), 2914-503 Setubal, Portugal; 4VALORIZA—Research Centre for Endogenous Resource Valorization, Polytechnic Institute of Portalegre, 7300-555 Portalegre, Portugal; antoniocalha@ipportalegre.pt; 5School of Education and Social Sciences, Polytechnic Institute of Portalegre, 7300-109 Portalegre, Portugal; 6Unidade de Investigação em Ciências Orais e Biomédicas (UICOB), Faculdade de Medicina Dentária, Universidade de Lisboa, Rua Teresa Ambrósio, 1600-277 Lisbon, Portugal; 7Center for Innovative Care and Health Technology (ciTechcare), Polytechnic of Leiria, 2410-541 Leiria, Portugal; 8Group Inovation and Development in Nursing (NursID), Centro de Investigação em Tecnologias e Serviços de Saúde (CINTESIS), 4200-450 Porto, Portugal; carlossequeira@esenf.pt; 9Nursing School of Porto, 4200-072 Porto, Portugal

**Keywords:** adolescents, mental health, mental health literacy, positive mental health, schools

## Abstract

This study aimed to assess adolescents′ Mental Health Literacy (MHL) level, Positive Mental Health (PMH) level, the association between sociodemographic variables and the MHL and PMH levels, and the relationship between adolescent′s MHL and PMH levels. A quantitative, cross-sectional, correlational study was conducted with a convenience sample of 260 adolescents studying in the 5th to 12th years of school. The Mental Health Knowledge Questionnaire, the Mental Health-Promoting Knowledge, and the Positive Mental Health Questionnaire were used for data collection. Most of the adolescents were female (55.8%) with a mean age of 14.07 years. The participants showed good levels of MHL (MHKQ). The participants showed good levels of MHL (MHKQ
$\bar{𝘹}$
= 60.03; MHPK-10 $\bar{𝘹}$ = 4.49) and high levels of PMH ($\bar{𝘹}$ = 128.25). The adolescents with higher levels of MHL were the oldest, in a higher year of school, female, those whose mothers are employed, those who have healthy eating habits, and those who have a better body image self-perception. Adolescents in a lower year of school, with adequate sleep habits, who spend fewer hours a day in front of a screen or online, and who have a better self-perception of mental and physical health and body image were the ones with higher PMH levels. These findings suggest the need to implement experimental or quasi-experimental studies to ascertain the effectiveness of interventions that promote adolescents′ positive mental health literacy.

## 1. Introduction

In recent years, adolescents′ mental health has increased experts′ concern, not only due to the significant prevalence of mental health disorders in this population [1], but also due to the early age of onset of the first episode of these disorders—before 14 years of age [2]. If, besides these factors, the profound changes that occur in adolescence at the physical and mental level [3] and this population′s low/modest levels of mental health literacy [3,4,5,6] are considered, then adolescents′ mental health vulnerability becomes evident.

Adolescence is a phase of life characterized by an important development in many levels: physical, psychological, emotional, and social. Defined by the World Health Organization (WHO) as the phase between childhood and adulthood, adolescence encompasses the period from 10 to 19 years old [7]. This wide range of time can be divided into stages, which may differ according to many authors. In our study, we adopted the division in three stages: early adolescence (10–14 years), middle adolescence (15–16 years), and late adolescence (17–19 years) [8].

Aware of the vulnerability of some groups in society, such as adolescents, the WHO [9] via their Mental Health Action Plan 2013–2020 defined that one of the goals for all nations is to implement strategies for mental health promotion and prevention, targeting all citizens, not just people with diagnosed mental disorders. This Action Plan was extended until 2030 by the WHO [10] and is aligned with the United Nations Sustainable Development Goals (SDG), namely with Goal 3–Good Health and Well-Being to “ensure healthy lives and promote well-being for all at all ages” [11].

In line with the WHO Mental Health Action Plan and the United Nations′ SDGs, there has been growing global research interest in the salutogenic dimension of mental health, Positive Mental Health (PMH), and Mental Health Literacy (MHL), both considered protective elements of mental health.

Conceptually speaking, Jorm et al. [12] defined MHL as the “knowledge and beliefs about mental disorders which aid their recognition, management or prevention” (p. 182). However, this concept has evolved and has become somewhat broader, encompassing not only knowledge but also skills to promote individual and community mental health [13], consisting of four components: understanding how to obtain and maintain good mental health, understanding mental disorders and their treatments, decreasing the stigma related to mental disorders, and increasing the effectiveness of help-seeking [14]. Therefore, MHL is a concept that implies the empowerment of the individuals to provide themselves with mental care [15] and the consequent empowerment of the community, which is an important strategy for promoting populational health and achieving health gains.

Concerning adolescents, several studies have shown that this population has low/moderate levels of MHL [3,4,5,6]. However, many of this research has focused mainly on the knowledge component of mental disorders, the stigma component and the help-seeking component, while few studies have addressed the knowledge component of how to obtain and maintain good mental health in this population [16,17]. Therefore, further studies on this component of MHL are needed to fill this research gap.

The concept of PMH emerges from the concept of mental health and has its origins in Seligman′s positive psychology and the work developed by Marie Jahoda in 1958 [18]. Although there is no universal definition, PMH is considered one of the dimensions of mental health and consists of a person′s ability to understand his/her environment and to adapt to it or modify it to strengthen his/her optimal functioning [19].

Based on this concept, Lluch-Canut developed the Multifactor Model of Positive Mental Health (MM-PMH) as an explanatory model of the PMH construct, composed of six interrelated factors: Personal Satisfaction (F1), Pro-Social Attitude (F2), Self-Control (F3), Autonomy (F4), Problem Solving and Self-Actualization (F5), and Interpersonal Relationship Skills (F6) [19,20]. In the MM-SMP, the indicators of F1–Personal Satisfaction are self-concept, self-esteem, satisfaction with personal life, and optimistic outlook on the future; the indicators of F2–Pro-Social Attitude are sensitivity of the person to the social environment, altruistic attitude of support for others, and acceptance of others and of different social facts; the indicators of F3–Self-Control are ability to cope with stress and problematic situations, emotional balance/control, and tolerance to stress, anxiety, and frustration; the indicators of F4–Autonomy are independence, the ability to self-regulate behavior, and self-confidence/personal security; the indicators of F5–Problem Solving and Self-Actualization are the ability to make decisions, analytical skills, adaptation to change, and an attitude of continuous personal development; finally, the indicators of F6–Interpersonal Relationship Skills are empathy, the ability to emotionally support others, and the capacity to establish interpersonal relationships globally and more intimately [19,20].

It is possible to assess PMH based on the six factors of the MM-PMH using the Positive Mental Health Questionnaire, also created by Lluch-Canut [18]. According to the score obtained in this questionnaire, a person is considered to be at a low, intermediate, or high level of PMH. The high level of PMH corresponds to a flourishing state, that is, a state in which the person has positive feelings and high levels of emotional, psychological, and social well-being combined with the absence of psychopathology [21], and the low level of PMH corresponds to a state of languishing, that is, to a state of absence of mental health [21].

PMH has been assessed in various settings using the Positive Mental Health Questionnaire, namely in people with chronic problems [22,23], persons with schizophrenia [24], university students [25,26], caregivers of people with schizophrenia [27], health professionals working in mental health services [28], and university teachers [29]. However, there is a gap in research in this area: the study of PMH in adolescents. As PMH is considered a catalyst of positive functioning and psychological well-being, there is a clear need to know the levels of PMH in adolescents due to its impact on this population′s current and future mental health.

To the best of the authors′ knowledge, the relationship between MHL and PMH in adolescents has not yet been studied. Therefore, carrying out such an investigation is critical.

Considering the above, the following research questions were formulated:(1)What is adolescents′ level of MHL?(2)What is adolescents′ level of PMH?(3)What is the association between sociodemographic variables and adolescents′ levels of MHL and PMH?(4)What is the relationship between adolescents′ level of MHL and their level of PMH?

## 2. Materials and Methods

### 2.1. Aims

This study is part of a broader research project on Literacy and Positive Mental Health of Adolescents, conducted by researchers J.N., C.S., and C.F.-G. In this article, the objectives were to assess adolescents′ level of mental health literacy; assess adolescents′ level of positive mental health; evaluate the relationship between variables of sociodemographic characterization and MHL and PMH levels; and correlate the level of PMH with adolescents′ level of MHL.

### 2.2. Study Design

A descriptive, cross-sectional, and correlational study was conducted from April to June 2020 using an online questionnaire available to the participants in the Google^®^ Forms platform (Google LLC, Mountain View, CA, USA). All collected data are confidential. Informed consent was obtained. The STROBE Statement checklist [30] was used to write this article.

### 2.3. Participants and Setting

The convenience sample included adolescents from the 2nd cycle, 3rd cycle, and secondary education (5th–12th year of school), from three urban public schools in a district of the North Alentejo region in Portugal.

Adolescents were eligible to answer the questionnaires if they were attending a school whose principal had authorized participation in the study, if their parents/legal representatives had given their permission via written informed consent, and if the adolescents were 10 to 19 years old. The exclusion criteria were adolescents with cognitive disorders, adolescents without informed consent given by their parents/legal representatives, and adolescents who refused to participate in the study.

The sample was obtained by the non-probability convenience sampling method. We calculated our sample for a margin of error of 5% and a reliability of 90%, which would result in a necessary sample of 242 participants. However, we accepted 18 more in case there were some null questionnaires. Therefore, from a population of 2125 adolescents, a total of 260 participants were recruited, after 13 duplicates were eliminated. With this number of adolescents in our sample, the probability was 90% that the study would detect a relationship between the independent and the dependent variables at a two-sided 0.05 significance level, if the true change in the dependent variables was 0.202 standard deviations per one standard deviation change in the independent variable, according to the MGH Biostatistics Center Sample Size Calculator [31]. All the participants who voluntarily completed the online questionnaires made the convenience sample.

The data collection took place from 17 April to 30 June 2020, in the 2nd cycle, 3rd cycle, and secondary schools (5th to 12th year of school) that agreed to participate in the study. An initial meeting was held with the schools′ principals to explain the research purpose and the data collection procedures. Initially, the plan was that parents/legal representatives fill in hard copies of the informed consent, and adolescents fill in hard copies of the questionnaires provided by a research team member in a classroom environment. However, due to the COVID-19 pandemic, the Portuguese Government suspended face-to-face teaching activities in schools on 16 March 2020. This change implied redefining the data collection procedure to be online. A member of the research team (J.N.) emailed the link of the informed consent to the principals of the schools, who in turn forwarded it to the adolescents′ parents/legal representatives. The link for the adolescents to complete the questionnaires was sent to the email provided by the parents/legal representatives who gave consent. The informed consent and the questionnaires were created in Google^®^ Forms (Google LLC, Mountain View, CA, USA). A document about the research study was attached to the email sent to the parents/legal representatives and the adolescents, explaining the methodology, the conditions and funding, the potential risks, the potential advantages, the confidentiality and anonymity of data, the way of disseminating the results, and the members of the research team and their contacts.

### 2.4. Data Sources and Measurement

The instruments used to evaluate the adolescents at a single time-point were the following: (1) Questionnaire for the Characterization of the Adolescents, (2) The Mental Health Knowledge Questionnaire [15,32], (3) The Mental Health-Promoting Knowledge [15,32], and (4) Positive Mental Health Questionnaire [33].

#### 2.4.1. Characterization of the Sample

The Questionnaire for the Characterization of the Adolescents was used in the sociodemographic characterization of the adolescents. This questionnaire was developed by the study′s research team and is composed of 29 self-report items: age; gender; year of school education; employment status and occupation of father and mother; history of mental health problems; previous contact with people with mental health problems; sleep habits; medication consumption; leisure and exercise activities; eating habits; internet and gadget use; interpersonal relationships; alcohol and tobacco consumption; self-perception of mental and physical health and body image; and self-perception of mental and physical health during confinement due to the COVID-19 pandemic. A pre-test of this questionnaire was conducted with 10 adolescents aged 10 to 17 years old. Minor changes were made according to their suggestions to make the characterization questionnaire more understandable and easier to complete for all adolescents.

#### 2.4.2. Mental Health Literacy

The researchers used two scales to assess all components of MHL: The Mental Health Knowledge Questionnaire (MHKQ) and The Mental Health-Promoting Knowledge (MHPK-10).

The MHKQ scale assesses the knowledge and awareness of mental health. This scale was translated and validated into Portuguese by Chaves et al. [15,32] based on the instrument developed by the Chinese Ministry of Health in 2009 [34]. The scale is composed of 20 self-report items, in which items 1–16 assess knowledge about the characteristics of mental health and mental disorders and belief in their epidemiology, while items 17–20 assess awareness of mental health promotion activities. Responses to items 1–16 are on a Likert scale of 1 to 5 points (1 = strongly disagree; 5 = strongly agree), and answers to items 17–20 are on a dichotomous scale of “yes” or “no” (yes = 1 point; no = 0 points). The MHKQ scale is divided into three MHL dimensions: dimension 1–knowledge of the characteristics of mental health and mental disorders (items 1, 2, 3, 5, 7, 8, 11, 12, 15, and 16), dimension 2–belief in the epidemiology of mental disorders (items 4, 6, 9, 10, 13, and 14), and dimension 3–awareness-raising on mental health promotion activities (items 17–20). Higher scores correspond to higher levels of mental health knowledge.

The Mental Health-Promoting Knowledge (MHPK-10) instrument was developed by Bjørnsen et al. [16] and translated to and validated for Portuguese by Chaves et al. [15,32]. This instrument presents the α Cronbach of 0,87 in the Portuguese version. The MHPK-10 evaluates the positive component of the MHL, that is, the level of knowledge about the factors that help obtain and maintain good mental health in the autonomy, relationship, and competence dimensions [17]. This scale consists of 10 self-report items, each rated on a Likert scale from 1 to 5 points (1 = strongly disagree to 5 = strongly agree; there is also the option “N/A” = not applicable, which corresponds to zero points). The higher the mean score, the higher the level of knowledge on the factors that help obtain and maintain good mental health. It should be noted that a mean score < 4 corresponds to an insufficient level of knowledge since values 4 and 5 correspond to the correct answers for each item [16].

#### 2.4.3. Positive Mental Health

The Positive Mental Health Questionnaire (PMHQ) was used to assess the participants’ levels of positive mental health. This questionnaire was developed by Lluch-Canut [18] and translated to and validated for the Portuguese population by Sequeira et al. [33] with a 0.92 α Cronbach. This questionnaire consists of 39 self-reported items, unevenly distributed by six factors: F1–Personal Satisfaction (items 4, 6, 7, 12, 14, 31, 38, and 39), F2–Pro-Social Attitude (items 1, 3, 23, 25, and 37), F3–Self-Control (items 2, 5, 21, 22, and 26), F4–Autonomy (items 10, 13, 19, 33, and 34), F5–Problem Solving and Self-Actualization (items 15, 16, 17, 27, 28, 29, 32, 35, and 36), F6–Interpersonal Relationship Skills (items 8, 9, 11, 18, 20, 24, and 30). The items of the PMHQ are stated in a positive or negative way. The answers to the statements are presented on a Likert-type scale from 1 to 4 points according to the frequency with which a given behavior occurs (1 = always or almost always; 2 = most of the time; 3 = sometimes; 4 = rarely or never). In the PMHQ, a score is obtained for each factor, and then a total score is measured that corresponds to the sum of all items. The score of the PMHQ ranges from 39 to 156 points (low level or languishing state–39 to 78 points; intermediate level–79 to 117 points; high level or flourishing state–118 to 156 points). The items that are positively worded were inverted so that both the total score and the score of each factor are directly proportional to the positive mental health [20].

### 2.5. Statistical Analysis

The statistical analysis was carried out using the software IBM^®^ SPSS^®^ version 27 (Statistical Package for the Social Sciences–IBM Corp, Armonk, NY, USA) for Windows.

Descriptive statistics were used to characterize the adolescent′s sample. Mean and standard deviation were calculated to describe quantitative variables, and absolute and relative frequencies were calculated to describe qualitative variables. Regarding the inferential statistics, after verifying that the distribution of the mean was not normal (Shapiro Wilks test *p* < 0.001), the Eta test was used to access the association between the sociodemographic nominal variables and the MHKQ, MHPK-10, and the PMHQ. The Spearman′s correlation coefficient was calculated to access the relation between the sociodemographic quantitative variables and the MHKQ, MHPK-10, and the PMHQ; to access the relationship between MHKQ, MHPK-10, and the PMHQ; and to access the relationship between MHKQ, MHPK-10, and each factor of the PMHQ. The Gamma test was calculated to access the association between MHKQ, MHPK-10, PMHQ, and the three stages of adolescence. A level of *p* < 0.05 was considered as the basis for statistical significance.

## 3. Results

### 3.1. Sample Characteristics

The convenience sample of our study consisted of 260 adolescents, aged 10 to 19 years old ($\bar{𝘹}$ = 14.07, SD = 1.95). Of those, 13.5% were studying in the second cycle of basic education (10.8% in 5th year, 2.7% in 6th year), 59.6% were attending the third cycle of basic education (16.2% in 7th year, 19.2% in 8th year, 24.2% in 9th year), 27% were attending secondary education (10.4% in 10th year, 10.8% in 11th year, 5.8% in 12th year of school), and most of them were female (55.8%). Regarding the parents′ professional situation, most were reported as employed (father 95.8%; mother 90.8%).

Regarding mental health, most adolescents reported having no mental health problems (98.5%), not having used a health service in the last three months due to a mental health problem (97.7%), not having been followed up by a psychologist/psychiatrist (69.6%), and not taking medication for a mental health problem (98.1%). When asked whether they had ever met someone with a mental health problem, most answered affirmatively (51.5%).

Concerning lifestyles, the adolescents in our sample reported to eat on average about four meals a day ($\bar{𝘹}$ = 4.33, SD = 0.94). Most of the participants reported that they eat fruit and vegetables daily (84.2%), and the majority reported practicing physical exercise regularly (73.1%) and sleeping on average about 8.5 h a day ($\bar{𝘹}$ = 8.45, SD = 1.19). In terms of substance use, the majority stated that they do not consume alcoholic drinks (90.8%) or smoke (96.5%). When asked about the number of hours in front of a screen, the participants reported an average of about 5.75 h a day (SD = 3.63) and about 5.60 h online per day (SD = 4.50).

Regarding social relationships, most adolescents said they had friends both at school (99.6%) and outside school (99.2%). When asked about violence, most adolescents replied that they had never been victims of violence (81.9%).

The adolescents in our sample self-perceived both their mental health ($\bar{𝘹}$ = 4.25, SD = 0.89) and their physical health ($\bar{𝘹}$ = 4.13, SD = 0.82) as “Good” and their body image as “Normal” ($\bar{𝘹}$ = 3.73, SD = 0.96). During the first confinement due to the COVID-19 pandemic, adolescents considered their physical health to be “Worse” ($\bar{𝘹}$ = 2.99, SD = 0.87) when compared to the period before the confinement and their mental health to be “The same” ($\bar{𝘹}$ = 3.07, SD = 0.67). The results related to the characterization of our sample are provided in Appendix A.

### 3.2. Descriptive Statistics for Mental Health Literacy

The mean score of the MHL–knowledge (measured with the MHKQ) of the adolescent′s sample was 62.03 (SD = 6.27, Minimum = 16, Maximum = 80), corresponding to a good level of MHL, as shown in Table 1. It should be noted that the items related to the beliefs in epidemiology of mental health problems were the ones with the lowest scores (Appendix A). Regarding the rate of awareness of mental health promotion activities, we found that most adolescents knew about “World Mental Health Day” (64.6%), but less than half knew about the “International Day for Suicide Prevention” (48.1%), the “International Day against Drug Abuse and Illicit Drug Trafficking” (37.7%), and “World Sleep Day” (29.6%) (Appendix A).

The results obtained with the MHPK-10 about the level of knowledge on the factors that help obtain and maintain good mental health revealed an overall mean value of 4.49 (SD = 0.66, Minimum = 0, Maximum = 5), with 15.8% of the adolescents showing an insufficient level of knowledge about these factors (i.e., they had a mean score < 4).

In a more detailed analysis by stage of adolescence, we found that the participants in early adolescence showed a lower level of MHL in MHKQ ($\bar{𝘹}$ = 61.5, SD = 6.08) compared with the other participants. We also found that the middle adolescence participants presented a lower level of MHL in MHPK-10 compared with the other stages (Table 2).

### 3.3. Descriptive Statistics for Positive Mental Health

As shown in Table 3, most of the adolescents presented a high level of PMH (78,5%), and the overall PMHQ mean score was 128.25 (SD = 14.71), placing the adolescents of our sample in the flourishing stage. In a more detailed analysis by factor of the PMHQ, F3–Self-Control and F4–Autonomy were the factors with the lowest mean values, at the intermediate level of positive mental health, indicating that adolescents have difficulty in the emotional control/emotional balance of negative emotions and thoughts and also indicating a low self-confidence level (Appendix A). On the other hand, F5–Problem-solving and Self-Actualization and F1–Personal Satisfaction presented the highest mean values, indicating that our adolescent sample reported the abilities to make decisions, to adapt to change, and to continuous growth/development and also reported self-esteem, satisfaction with personal life, and optimistic outlook on the future (Appendix A).

In a more detailed analysis by stage of adolescence, we found that the participants of the three stages showed a high level of PMH; however in the late adolescence group, the participants were the ones who presented the lowest level of PMH. By analyzing each factor of PMH, we found that, in late adolescence, the participants had the lowest level of PMH in F1–Personal Satisfaction, F2–Pro-Social Attitude, F3–Self-Control and in F4–Autonomy; in middle adolescence, the F5–Problem Solving and Self-Actualization was the group with the lowest level of PMH, and in early adolescence, the F6–Interpersonal Relationship Skills had the lowest level of PMH (Table 4).

### 3.4. Relationship between MHL and Sociodemographic Data

Table 5 shows the inferential statistics performed to investigate the relationship between the sociodemographic variables and MHL. As previously mentioned, to assess MHL, we used two scales (the MHKQ and the MHPK-10), and we determined the relationship between the sociodemographic variables and both scales.

Using the Eta test, it was possible to identify a statistically significant association with a weak negative Spearman correlation between MHL and mother having a job (Eta = 0.028, r_s_ (260) = −0.046, *p* = 0.465), recourse to a health service due to a mental problem in the last three months (Eta < 001, r_s_ (260) = −0.005, *p* = 0.932), daily fruit/vegetables intake (Eta = 0.022, r_s_ (260) = 0.047, *p* = 0.450), consumption of alcoholic beverages (Eta = 0.012, r_s_ (260) = −0.009, *p* = 0.879), and tobacco consumption (MHKQ: Eta = 0.046, r_s_ (260) = −0.057, *p* = 0.364; MHPK-10: Eta = 0.029, r_s_ (260) = −0.057, *p* = 0.357). We also identified a statistically significant association with a weak positive Spearman correlation between MHL and sex (Eta = 0.041, r_s_ (260) = 0.021, *p* = 0.731), psychologist or psychiatrist monitoring (Eta = 0.022, r_s_ (260) = 0.067, *p* = 0.283), previous contact with someone with a mental health problem (Eta = 0.006, r_s_ (260) = 0.027, *p* = 0.664), and taking medication for a mental health problem (MHKQ: Eta = 0.010, r_s_ (260) = 0.024, *p* = 0.703; MHPK-10: Eta = 0.040, r_s_ (260) = 0.064, *p* = 0.301)). It should be noted that, regarding the variable having a mental health problem, we identified a statistically significant association with MHL; however, it presented a weak negative correlation with MHL in the MHKQ (Eta = 0.019, r_s_ (260) = −0.016, *p* = 0.794) and a weak positive correlation in the MHPK-10 (Eta = 0.040, r_s_ (260) = 0.073, *p* = 0.239).

By performing the Spearman′s coefficient test, we identified a statistically significant positive yet weak relationship between MHL and age (r_s_ (260) = 0.187, *p* = 0.003), year of school (r_s_ (260) = 0.198, *p* = 0.001), number of meals a day (MHKQ: r_s_ (260) = 0.133, *p* = 0.032; MHPK-10: r_s_ (260) = 0.159, *p* = 0.010), and body image self-perception (r_s_ (260) = 0.124, *p* = 0.045).

Using the Gamma test, we found that there was a statistically significant association between the MHKQ and the stage of adolescence (Gamma = 0.143, *p* = 0.029), with a weak positive Spearman′s correlation (r_s_ (260) = 0.130, *p* = 0.036); however, there was no association between MHPK-10 and the stages of adolescence (Gamma = −0.018, *p* = 0.799), as shown in Table 6.

There were no significant differences or relationships between MHL and the other sociodemographic variables.

### 3.5. Relationship between PMH and Sociodemographic Data

Table 5 also shows the inferential statistics obtained from the comparisons between PMH and the sociodemographic variables. The evaluation using the Eta test identified a statistically significant association, with a negative weak Spearman′s correlation between PMH and mother having a job (Eta = 0.017, r_s_ (260) = −0.020, *p* = 0.751).

With Spearman′s coefficient test, it was possible to identify a statistically significant negative weak relationship between PMH and year of school (r_s_ (260) = −0.126, *p* = 0.043), number of online hours a day (r_s_ (260) = −0.122, *p* = 0.049), and number of hours in front of a screen per day (r_s_ (260) = −0.137, *p* = 0.027). Using the same nonparametric test, we identified a statistically significant positive weak relationship between PMH and number of hours of daily sleep (r_s_ (260) = 0.226, *p* <0.001), self-perception of physical health (r_s_ (260) = 0.398, *p* < 0.001), and self-perception of mental health during the first outbreak in the COVID-19 pandemic (r_s_ (260) = 0.164, *p* = 0.048). We also identified a moderate positive statistically significant relationship between PMH and self-perception of mental health (r_s_ (260) = 0.414, *p* < 0.001) and self-perception of body image (r_s_ (260) = 0.411, *p* < 0.001).

As shown in Table 6, using the Gamma test, we found that there was no association between PMHQ and the stages of adolescence (Gamma = −0.111, *p* = 0.101).

No significant differences or relationships were identified between the level of PMH and the other sociodemographic variables.

### 3.6. Correlations between PMH and MHL

The results of the relationship between MHL and PMH are presented in Table 7.

The relationship between mental health knowledge (measured with the MHKQ) and positive mental health (measured with the PMHQ) was investigated using Spearman′s correlation coefficient. We identified a significant and positive but weak relationship between the two variables (r_s_ (260) = 0.157, *p* = 0.011), with high MHL levels associated with high PMH levels. In a more detailed analysis per factor of the PMH scale, we found a weak positive significant relationship between the mental health knowledge and F2–Pro-Social (r_s_ (260) = 0.153, *p* = 0.016), F5–Problem-Solving and Self-Actualization (r_s_ (260) = 0.205, *p* = 0.001), and F6–Interpersonal Relationship Skills (r_s_ (260) = 0.195, *p* = 0.002).

The relationship between knowledge about the factors that help obtain and maintain good mental health (measured with the MHPK-10) and positive mental health (measured with the PMHQ) was also assessed using Spearman′s correlation coefficient. We identified a significant positive relationship, albeit weak but almost moderate, between the two variables (r_s_ (260) = 0.292, *p* < 0.001), with high levels of knowledge associated with high levels of PMH. By performing a more detailed analysis per factor of the PMH scale, we found a significant positive but weak relationship between knowledge about the factors that help obtain and maintain good mental health and all the factors of the PMH scale.

## 4. Discussion

The first research question of this study aimed to assess the adolescents′ level of MHL. The results obtained show that the adolescents had a good overall level of MHL, which is in line with some previous studies [3,5,16]. These results may be associated with the progressive focus on mental health that has existed worldwide in recent years [9,10,11]. It was interesting to note that the components of the MHL with the best results were those related to help-seeking and knowledge about how to obtain and maintain good mental health, which is in line with the results obtained by other researchers [3,5,16,34]. On the other hand, the component related to knowledge about mental disorders, their etiology, and their treatments was the one that presented the lowest scores [34]. In addition, we also found that, similar to the study by Yu et al. [34], adolescents showed a lack of awareness of mental health-promoting activities, in the case of our research specifically about “World Sleep Day” and the “International Day against Drug Abuse and Illicit Drug Trafficking”. This lack of awareness can be attributed to the fact that, in recent years, the Portuguese Government has focused on raising the overall awareness of Mental Health and not so much on specific problems. It became evident that the results arising from the MHKQ and the MHPK-10 suggest the need to focus on interventions promoting MHL both at the level of knowledge of mental disorders, their etiology, and their treatments and the level of awareness of mental health-promoting activities to avoid misconceptions about mental health and the consequent possible negative attitudes towards people with mental health problems. The results also suggests that it is important to begin the promotion of MHL in the early stages of adolescence because those adolescents had the lowest levels. This way, we are contributing to reducing the appearance of mental health disorder episodes before the age of 14 [2].

The second research question aimed to assess the adolescents′ levels of positive mental health. The results obtained indicate a high level of PMH of the adolescents in our sample, positioning them in the flourishing state. These results are higher than those of the study by Alves [35], with adolescents in the third cycle of basic education (7th to 9th years of school), and the study by Sequeira et al. [26], with nursing university students, that obtained intermediate levels of PMH, but they are in line with the results of Garcia′s study [36] and the Teixeira et al. study [37]. In our study sample and by considering the various and rapid physical, psychological, and social changes that occur at this stage of the life cycle, we obtained a higher level of PMH than we expected. We believe that this high level of PMH may be because the adolescents in our sample belong to schools that have had projects in partnership with the Primary Health Centre to develop socio-emotional skills since 2015, implemented by teams composed of nurses and psychologists.

Through a more detailed analysis per factor of the scale, we found that adolescents had intermediate levels of PMH regarding Self-Control (Factor 3) and Autonomy (Factor 4) and have difficulties in emotional control/emotional balance, especially because of negative emotions and thoughts and a low level of self-confidence. On the other hand, we found that the adolescents in our sample had higher levels of PMH in Problem-Solving and Self-Actualization (Factor 5) and also in Personal Satisfaction (Factor 1), showing capacity for decision-making, problem-solving, adaptation to change, as well as satisfaction with themselves, their personal life, and optimism for the future. Despite the good results in both factors (Factor 5 and Factor 1), the adolescents in our sample can still improve their personal development attitudes and self-esteem. These results differ somewhat from those obtained by Garcia [36], Alves [35], and Sequeira et al. [26] but are perfectly aligned with the specific characteristics of adolescents widely described in the literature. That is, the adolescents in our sample had a mean age of 14.07 years old, which places them in the end of early adolescence [8,38]; thus, it makes sense that our sample has not yet demonstrated high levels of self-control and autonomy, since at this stage of adolescent development they still resort to concrete thinking in stressful situations and have a vision of life and the world based on impossible dreams and illusion [8,38]. Thus, it is during this stage of the life cycle that adolescents learn to deal with impulses and emotions, that they begin to know their *self*, and that they start to gain the self-confidence necessary to be more autonomous [8,38]. It also makes sense that the adolescents in our sample had high levels of Personal Satisfaction and Problem-Solving and Self-Actualization skills, since they are in a period characterized by the brain growth [8,38], which provides, at this stage, that the development of new cognitive skills takes place, abstract reasoning and more realistic decision-making come together, and more stable emotional, sexual, and social bonds are established [8,38].

The third research question of our study sought to determine the relationship between sociodemographic variables and the levels of MHL and PMH. The results of our study indicate that the adolescents who had higher levels of MHL were the oldest; those who were in a higher year of school; those whose mothers were employed; those who ate a greater number of meals a day; the girls; those who ate fruit and vegetables daily; those who had a better self-perception of their body image; those who had no previous contact with someone with a mental health problem; and those who had not been monitored by a psychologist or a psychiatrist. Therefore, we found that, as age increased, the level of MHL also increased, being the adolescents who were in the stage of late adolescence had higher levels of MHL, which is in line with the results of Campos et al. [4] but contradicts those of Yu et al. [34]. This points to the importance of implementing, in the future, interventions that promote MHL in the early stages of adolescence. We also found that the higher the year of school, the higher the level of MHL, which may be attributed to the rapid changes in cognitive and emotional development that occur during adolescence, which is in line with the findings of Campos et al. [5]. The relationship between the adolescents′ MHL and the fact that the mother had a job may be associated with the adolescents′ perception of a good financial situation, enabling them to use mental health services and participate in mental health-promoting activities, which often have restricted access because they require payment. Other studies have also found a positive relationship between MHL and perceived good financial position [17] or a good socio-economic status [3] and a negative relationship between MHL and financial disadvantage [39]. Interestingly, in our sample of adolescents, we found a positive relationship between MHL and the number of meals per day and the daily consumption of fruit and vegetables. The explanation we found is the fact that these two variables are usually associated with healthier eating habits and, consequently, with healthier lifestyles, and in the existing literature, other studies have also shown that people with high levels of MHL enjoy better physical and social health, since they have a high sense of seeking information about health and well-being, as well as the ability to understand and use this information to change unhealthy behaviors or to maintain a healthy lifestyle, which logically can lead to good general health outcomes [39,40]. The same justification can be applied to explain the positive relationship between MHL and the self-perception of body image, since it is during adolescence that the creation and development of the body image occurs [8,38]. As shown in previous studies of other researchers, the results of our study indicate that girls presented higher levels of MHL [4,5,17]. The relationship found between the highest levels of MHL among adolescents and those who had not been monitored by a psychologist or a psychiatrist can be explained by the fact that a person usually turns to those health professionals when he/she have a mental health problem and do not know how to deal with/solve it. In this case, apparently the person is literate when it comes to seeking help but is not literate about how to obtain or maintain good mental health. These results require further clarification to gain a better understanding of the relationship between MHL and having therapist sessions. In our study, we obtained a negative influence of the variable contact with someone with a mental health problem on MHL, which contradicted the results of the studies of Campus et al. [3] and Campus et al. [4].

Concerning PMH, the results of our study showed that the adolescents with the highest levels of PMH were those who sleep the most hours a day, those who spend fewer hours a day in front of a screen, those who spend the fewest hours online each day, those studying in the lower years of school, those who seem to have a better self-perception of both their MH, PH and body image, and lastly, and those who seem to have had a better self-perception of their MH during the confinement due to the COVID-19 pandemic. The positive relationship between the number of hours of sleep per day and PMH is in line with the results of Guo et al. [41] and is explained by the association between sleeping the recommended number of hours and improved memory, attention, learning, behavior, emotional regulation, quality of life, and, therefore, better mental and physical health [36,42]. We also found that the number of daily hours in front of a screen and the number of hours online per day had a negative relationship with the levels of PMH, which confirms what some existing literature had already told us, that is, that the use of screens for a moderate or long time throughout the day is associated with low psychological well-being, namely low self-control, less emotional stability, less concentration, inability to finish tasks, and less ability to make friends [36,43,44]. This situation is particularly worrying in adolescents, as they are at a stage of important cognitive, psychological, emotional, and social development [8,38], and the harmful effects of excessive screen use may leave negative marks that will last into adulthood [45]. However, as we currently live in a digital age, it is necessary to find strategies, together with adolescents, that minimize these negative consequences and turn them into opportunities for their development. It was also interesting to note that, as the year of school increased, the PMH of the adolescents in our sample decreased. This relation can be attributed to the increasing academic demands placed on adolescents as they approach the end of secondary school, to making increasingly difficult decisions related to the future (in academic, professional, emotional, and social terms), to a greater identification with the peer group, and, possibly, to the consequent greater predisposition to peer group influence/pressure in adolescents′ lives [8,38]; all this is very characteristic of this stage of development [8,38]. For all these reasons, it is important that, in future studies, this possible relationship continues to be investigated to obtain a more solid basis of scientific evidence. Finally, the results obtained show the existence of a positive relationship between PMH and self-perception of PH, MH and body image, possibly associated with the fact that our sample consists of adolescents who mostly adopt healthy lifestyles (such as regular physical activity, low consumption of tobacco and alcohol, eating healthy food five to six times a day, in line with the definition of the World Health Organization [46]), which leads to increased feelings of psychological well-being [47] and better mental health [48], consequently leading to a better self-perception of their health, which produces very beneficial effects on the adolescents′ levels of PMH.

Finally, the fourth research question of our study aimed to correlate the level of PMH with the adolescents′ level of MHL. The results in our study showed that MHL levels had a positive relationship with PMH levels, showing that, as literacy increased, so did PMH. In this regard, it is worth noting the significant and positive relationship between knowledge about obtaining and maintaining good mental health and all the PMH factors, confirming the research team′s expectations. As the present study was the first to correlate MHL and PMH among adolescents, our results cannot be compared with other studies, although we can mention that our results are in line with those obtained in the recently published study by Teixeira et al. [37] among University students. The finding of this relationship provides a huge motivation for implementing interventions that improve adolescents’ MHL levels to simultaneously obtain high levels of PMH in adolescents.

Despite this study′s scientific and methodological rigor, some limitations should be reported and taken into consideration when interpreting the results. First, as it is an investigation involving the participants′ self-report, we should consider the risk of bias in the responses influenced by social desirability, making it very difficult to interpret the adolescents′ behavior from their self-reports. Second, the limitation related to the type of study carried out (cross-sectional study) did not allow for the assessment over time of the levels of MHL or PMH, so the causes and effects of the relationships between the variables could not be determined. Lastly, we had a convenience sample, which does not allow the generalization of the results obtained.

## 5. Conclusions

This correlational cross-sectional study assessed the level of adolescents′ MHL, PMH, and their relationship with sociodemographic variables and explored the relationship between MHL and PMH.

The adolescents in our study showed good levels of MHL and high levels of PMH. Positive significant associations/relationships were observed between MHL and age, year of school, number of daily meals, mother′s employment status, female sex, daily intake of fruit and vegetables, and good self-perception of body image. Significant positive associations/relationships were also observed between PMH and number of hours of sleep and self-perception of PH, MH, and body image. In addition, significant negative associations/relationships were found between PMH and adolescents′ years of school and the number of daily hours online and in front of a screen. Finally, a positive correlation was found between the MHL and PMH of the adolescents.

We believe that, in the future, we should focus on carrying out longitudinal studies to understand if the levels of MHL and PMH are maintained over time. In addition, we should carry out more studies to determine if there are differences between the stages of adolescence and carry out quasi-experimental or experimental community studies of interventions promoting adolescents′ positive mental health literacy, especially targeting the early adolescence stage. These studies will make a huge contribution so that in the future we may have positive, mentally healthy adults.

## Figures and Tables

**Table 1 ijerph-19-08165-t001:** Mental Health Literacy descriptive statistics (*n* = 260).

	*N*	Min.	Max.	Mean	SD
MHKQ–Knowledge	260	16	80	62.03	6.27
MHPK-10	260	0	5	4.49	0.66

Abbreviations: Max., maximum; Min., minimum; *n*, number of cases; SD, standard deviation.

**Table 2 ijerph-19-08165-t002:** Levels of MHL according to the stage of adolescence (*n* = 260).

Stages of Adolescence	*n*	MHKQ				MHPK-10			
		Min.	Max.	Mean	SD	Min.	Max.	Mean	SD
Early adolescence	153	43	73	61.5	6.08	1.30	5	4.49	0.64
Middle adolescence	75	45	78	62.42	6.86	0	5	4.43	0.77
Late adolescence	32	44	72	63.59	5.48	3.20	50	4.55	0.51

Abbreviations: Max., maximum; Min., minimum; MHL, mental health literacy; *n*, number of cases; SD, standard deviation.

**Table 3 ijerph-19-08165-t003:** Positive Mental Health Levels, global and per factor (*n* = 260).

	*n*	%	Mean	SD	Levels of PMH		
					Low	Middle	High
PMHQ Global Score			128.25	14.71	39–78	79–117	117–156
PMHQ Factors:							
F1. Personal Satisfaction			27.45	4.22	8–16	17–24	25–32
F2. Pro-social Attitude			17.87	2.21	5–10	11–15	16–20
F3. Self-Control			14.96	3.20	5–10	11–15	16–20
F4. Autonomy			15.69	3.01	5–10	11–15	16–20
F5. Problem-Solving and Self-Actualization			29.32	4.46	8–18	19–27	28–36
F6. Interpersonal Relationship Skills			22.97	3.58	7–14	15–21	22–28
PMH Global Levels:							
Low level (languishing)	1	0.4					
Intermediate level	55	21.2					
High level (flourishing)	204	78.5					
Total	260	100					

Abbreviations: *n*, number of cases; PMH, positive mental health; SD, standard deviation.

**Table 4 ijerph-19-08165-t004:** Levels of PHL according to the stage of adolescence (*n* = 260).

Stages of Adolescence	*n*	PMHQ Total Score		PMHQ F1		PMHQ F2		PMHQ F3		PMHQ F4		PMHQ F5		PMHQ F6	
		Mean	SD	Mean	SD	Mean	SD	Mean	SD	Mean	SD	Mean	SD	Mean	SD
Early adolescence	153	129.26	14.55	28.05	3.99	17.89	2.09	15.05	3.12	15.93	2.93	29.46	4.26	22.89	3.68
Middle adolescence	75	127.24	14.89	26.96	4.23	17.95	2.40	15.08	3.19	15.37	3.08	28.96	4.99	22.92	3.54
Late adolescence	32	125.75	15.07	25.69	4.69	17.60	2.34	14.25	3.55	15.28	3.17	29.50	4.19	23.44	3.30

Abbreviations: *n*, number of cases; PML, positive mental health; SD, standard deviation.

**Table 5 ijerph-19-08165-t005:** Relationship between the sociodemographic characteristics and MHKQ, MHPK-10, and PMHQ (*n* = 260).

Sociodemographic Characteristics	*n*	MHKQ			MHPK-10			PMHQ		
		Eta	r_s_	*p*	Eta	r_s_	*p*	Eta	r_s_	*p*
Age	260		0.187	0.003		−0.002	0.975		−0.117	0.057
Sex	260	0.082			0.041 *	0.021	0.731	0.073		
Year of school	260		0.198	0.001		−0.039	0.530		−0.126	0.043
Employed father	260	0.093			0.111			0.053		
Employed mother	260	0.139			0.028 *	−0.046	0.465	0.017 *	−0.020	0.751
MH problem	260	0.019 *	−0.016	0.794	0.040 *	0.073	0.239	0.062		
Recourse to a health service due to MH problem (last 3 months)	260	<0.001	−0.005	0.932	0.071			0.123		
Psychologist or psychiatrist monitoring	260	0.083			0.022 *	0.067	0.283	0.257		
Contact with someone with a MH problem	260	0.169			0.006 *	0.027	0.664	0.115		
Hours of sleep/day	260		−0.050	0.419		0.082	0.188		0.226	<0.001
Taking medication for MH	260	0.010 *	0.024	0.703	0.044	0.064	0.301	0.060		
Regular exercise	260	0.084			0.101			0.291		
No. of meals/day	260		0.133	0.032		0.159	0.010		0.106	0.087
Daily fruit/vegetable intake	260	0.054			0.022 *	−0.047	0.450	0.193		
Hours online/day	260		0.015	0.805		0.020	0.746		−0.122	0.049
Hours in front of a screen/day	260		0.063	0.309		0.052	0.401		−0.137	0.027
Victim of violence	260	0.078			0.071			0.228		
Consumption of alcoholic beverages	260	0.012 *	−0.009	0.879	0.054			0.080		
Tobacco consumption	260	0.046 *	−0.057	0.364	0.029 *	−0.057	0.357	0.151		
MH self-perception	260		−0.028	0.652		0.106	0.087		0.414	<0.001
PH self-perception	260		0.052	0.406		0.096	0.122		0.398	<0.001
Body image self-perception	260		−0.035	0.579		0.124	0.045		0.411	<0.001
PH self-perception in outbreak	260		−0.044	0.478		−0.079	0.205		0.027	0.663
MH self-perception in outbreak	260		−0.053	0.398		−0.035	0.580		0.164	0.008

Abbreviations: MH, mental health; MHPK-10, Mental Health-Promoting Knowledge; MHKQ, Mental Health Knowledge Questionnaire; *n*, number of cases; No., number; *p*, significance value; PH, physical health; PMHQ, Positive Mental Health Questionnaire; r_s_, Spearman correlation value; *, significance statistic *p* value < 0.05.

**Table 6 ijerph-19-08165-t006:** Association between MHL and PML and the stage of adolescence (*n* = 260).

	*n*	MHKQ				MHPK-10				PMHQ			
		Gamma	*p*	r_s_	*p*	Gamma	*p*	r_s_	*p*	Gamma	*p*	r_s_	*p*
Stages of Adolescence	260	0.143	0.024 *	0.130	0.036	−0.018	0.799			−0.111	0.101		

Abbreviations: MHL, mental health literacy; *n*, number of cases; PML, positive mental health; *p*, significance value; r_s_, Spearman correlation value; SD, standard deviation; *, significance statistic *p* value < 0.05.

**Table 7 ijerph-19-08165-t007:** Correlation between MHKQ, MHPK-10, and PMHQ (*n* = 260).

	PMHQ Total Score		PMHQ F1		PMHQ F2		PMHQ F3		PMHQ F4		PMHQ F5		PMHQ F6	
	r_s_	*p*	r_s_	*p*	r_s_	*p*	r_s_	*p*	r_s_	*p*	r_s_	*p*	r_s_	*p*
MHKQ Total Score	0.157	0.011	0.063	0.314	0.153	0.016	0.031	0.622	0.005	0.939	0.205	0.001	0.195	0.002
MHPK-10 Total Score	0.292	<0.001	0.201	0.001	0.163	0.008	0.161	0.009	0.233	<0.001	0.232	<0.001	0.255	<0.001

Abbreviations: *p*, significance value; r_s_, Spearman correlation value; SD, standard deviation.

## Data Availability

Data available on request due to ethical restrictions.

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
