# Peer review of "Mental Health Literacy and Positive Mental Health in Adolescents: A Correlational Study"

_ijerph, 2022, doi:10.3390/ijerph19138165_

Round 1
Reviewer 1 Report
New version (follow up)
I read the new version but I can’t find the Authors’ reply.
This new version does not show any significant improvement with regards to the previous one. The analysis is mainly focused on the relationship between Mental Health Literacy and Positive Mental Health using a sample of Portuguese university students.
In my opinion the main problem is represented by the statistical analysis that is still weak and based on few observations - just 260.
Author Response
Dear Reviewer,
Please find our reply in the document attached.
Kind regards,
Joana Nobre
On behalf of all authors

Reviewer 2 Report
Authors made significant corrections
Author Response

(The authors gave the same response as above.)

Reviewer 3 Report
Revise paragraph 46 as it has been cut off and does not make sense.
Author Response

(The authors gave the same response as above.)

Round 2
Reviewer 1 Report
The authors worked intensively to improve their research. The main critical issues are not completely solved. Anyway I think the paper could be accepted in this final version.
This manuscript is a resubmission of an earlier submission. The following is a list of the peer review reports and author responses from that submission.
Round 1
Reviewer 1 Report
Authors prudently conducted a research on an interesting subject.
Discuss the developmental stages regarding the results obtained. 10-19 years is a very wide range including children prepuberts puberts.
The brain development physiology regardding the results should also be discussed.
Tables: decimals should be marked either with comma or point and the chosen form should be kept in all tables.
Reviewer 2 Report
Mental Health Literacy and Positive Mental Health in Adolescents: A Cross-sectional Study
The authors investigate Mental Health Literacy, Positive Mental Health and their relationship for a sample of Portuguese adolescents. Their main findings show that the sample of adolescents has a high level of MHL and PMH. MHL and PMH have a positive correlation.
Key comments
- The abstract should be more focused on the most important elements of the research.
- The data description is poorly presented.
- The statistical analysis should be described more in depth.
- Some findings are based on misleading assumptions (e.g. a higher level of MHL is associated to the number of meals per day. The authors assume that the number of meals affects MHL. In my opinion the key point is: What is the determining factor? Is MHL the determinant of the number of meals or vice versa?). The statistical analysis is very basic and it does not allow to disentangle this point and similar other points.
Moreover, the authors affirm that:
“Making adolescents aware of certain commemorative days in this area may be an important ally for reflection on the subject, thus contributing to an increase in adolescents' MHL.”
I think the authors should explain better the rationale of their assumption (for example a more in depth analysis of the previous literature could help).
Lines 452-454: “We found that as age increases, the level of MHL also increases, probably because older adolescents, in general, have greater skills for searching for information than younger adolescents and have a greater ability to understand mental health information.”
I have many doubts about this statement. For example the older adolescents could be simply more confident about their skills and this could improve their MHL.
Reviewer 3 Report
Dear authors, the subject matter is adequate and the article is interesting, but certain improvements are required.
On the one hand, with respect to the substantiation, it should be updated since there are numerous references of an invalid range since they are prior to the last 5 years, so although there may be old citations, they are too many, it is recommended to update them with ideas more appropriate to a topic as researched as mental health.
Examples of citations to update:
Kieling, C.; Baker-Henningham, H.; Belfer, M.; Conti, G.; Ertem, I.; Omigbodun, O.; Rohde, L. A.; Srinath, S.; Ulkuer, N.; Rah- 602
man, A. Child and Adolescent Mental Health Worldwide: Evidence for Action. Lancet 2011, 378 (9801), 1515-1525. 603
https://doi.org/10.1016/S0140-6736(11)60827-1.
Gulliver, A.; Griffiths, K. M.; Christensen, H. Perceived Barriers and Facilitators to Mental Health Help-Seeking in Young Peo- 622
ple: A Systematic Review. BMC Psychiatry 2010, 10, 113. https://doi.org/10.1186/1471-244X-10-113.
Etc.
Regarding methodology.
I believe that although justified, the sample is very small and further clarification is needed for a parametric study. The established indices are requested.
Likewise, the questionnaires are justified by blocks, so I do not understand why then tables are developed by items. Analyses should be carried out by summation, i.e. by the blocks that have been established. The tables and the results should be modified.
I do not enter into the assessment of the discussion until the statistics by blocks are performed and the rationale is updated.